# The impact pathway of new quality productive forces on regional green technology innovation: A spatial mediation effect based on intellectual property protection

**Kun Cheng[1], Jianhui Yin[ID][2]\*, Feiyan Wang[3], Min Wang[3]**

**1** The 43rd Research Institute of China Electronics Technology Group Corporation, Hefei, China, **2** East University of Heilongjiang, Harbin, China, **3** Anhui Business and Technology College, Hefei, China

\* 23002005@hljeu.edu.cn

## Abstract

Against the backdrop of global economic transformation and sustainable development, green technological innovation has become a core driver for enhancing national competitiveness and addressing environmental challenges. With the profound changes in production methods and technological innovation, new quality productive forces (NQPF) have emerged as a key driver of green technological innovation. This study aims to explore the mechanisms through which NQPF influence Regional Green Technology Innovation (RGTI), with a particular focus on the mediating role of intellectual property protection (IPP). Using panel data from 31 provinces in China from 2011 to 2022, we conduct an empirical analysis employing a spatial Durbin model and a spatial mediation effect model. The results indicate that NQPF significantly promote RGTI, particularly in enhancing resource utilization efficiency and greening production processes. However, the study also identifies an inverted U-shaped relationship between NQPF and green technological innovation, primarily driven by local dynamics, where the positive effect diminishes after reaching a certain threshold. Further analysis reveals that IPP plays a crucial mediating role, not only directly fostering green innovation but also amplifying the positive effects of NQPF by enhancing the efficiency of innovation outcomes. Based on these findings, this study offers policy recommendations for promoting RGTI, emphasizing the need to strengthen support for NQPF, improve IPP mechanisms, and build a regional collaborative innovation system.

## 1. Introduction

Amid the global economic transformation and the pursuit of sustainable development, green technological innovation has emerged as a core driver for enhancing national competitiveness and addressing environmental challenges. In recent years, with profound changes in production methods and technological advancements, NQPF have gained increasing attention in both academic and policy circles as a key driver of green technological innovation [1].

**Data availability statement:** All relevant data are within the paper and its Supporting information files.

**Funding:** This work was supported by the Major Research Projects of Humanities and Social Sciences in Universities in Anhui Province of China (2024AH040285 to FW) and General Topics for the 2024 Planning Project of the Chinese Society of Business Statistics (2024STY134 to JY).

However, existing research primarily focuses on the relationship between traditional productivity and innovation capacity [2,3], leaving a gap in the systematic study of the concept of NQPF and its impact on RGTI. NQPF, as an emerging form of productivity, emphasize not only resource efficiency and technological progress but also the realization of economic growth within a low-carbon, environmentally friendly, and sustainable development framework [4]. Unlike traditional productivity, the core of NQPF lies in enhancing green economic development through technological innovation, the accumulation of knowledge capital, and the upgrading of production methods [5]. Scholars generally agree that NQPF can significantly promote RGTI by optimizing resource allocation and improving production efficiency [6]. However, the literature lacks systematic empirical studies on how NQPF specifically influence regional green innovation, particularly regarding its mechanisms across different regions. Therefore, exploring the key pathways through which NQPF drive RGTI, especially its cross-regional spillover effects, holds significant theoretical and practical value.

Over the past decade, China has witnessed significant dynamic changes in the development of green technology innovation and NQPF. Specifically, from 2011 to 2022, the number of green patent applications nationwide experienced steady growth, with the eastern developed regions demonstrating a marked advantage in both the quantity and quality of green technology innovation [7]. Simultaneously, the NQPF index exhibited an overall upward trend, albeit with notable regional disparities. The eastern coastal areas, leveraging robust economic foundations and technological advantages, achieved rapid advancements in NQPF, whereas the central and western regions showed comparatively slower progress. Furthermore, the intensity of intellectual property protection strengthened significantly during this period, particularly in the field of green technology patents, providing a solid institutional foundation for the dissemination and commercialization of innovation outcomes [8]. These dynamic changes not only reflect China's concerted efforts to promote green technology innovation but also highlight the persistent regional imbalances in development, thereby offering essential contextual support for exploring the relationship between NQPF and regional green technology innovation in this study.

At the same time, IPP, as an important institutional arrangement for safeguarding technological innovation and economic development, is increasingly being incorporated into the research framework of regional innovation systems [9]. IPP not only provides the necessary legal safeguards for the technological innovation of NQPF but also, to a certain extent, determines the scope and effectiveness of the diffusion of innovation outcomes [10]. The relationship among NQPF, IPP, and RGTI has become a key topic in current research. Some scholars have proposed that NQPF, by strengthening IPP mechanisms, not only directly promote green technological innovation but also indirectly facilitate the diffusion and upgrading of green technologies across regions by enhancing the efficiency of innovation outcomes [11,12]. However, empirical research on the relationship between these three factors remains relatively scarce, particularly regarding the use of spatial econometric models to reveal the interactive mechanisms between NQPF, IPP, and RGTI, where no unified understanding has yet been established.

Building on the above discussion, this study examines the relationship between NQPF, RGTI, and IPP across 31 provinces in China. The primary aim is to use a SDM based on spatial panel data to assess the spatial effects of NQPF on RGTI from 2011 to 2022. Additionally, the novelty of this research lies in its analysis of the spatial mediation effect. Therefore, this study also seeks to investigate whether the increased strength of IPP moderates the relationship between NQPF and RGTI. Specifically, we aim to answer the following questions: Can the development of NQPF significantly influence RGTI, and what is the nature of this relationship? Furthermore, does IPP mediate the relationship between the development of NQPF and RGTI across the 31 provinces in China?

The primary contributions of this paper are as follows: First, this study innovatively incorporates NQPF into the research framework of RGTI, systematically exploring the impact pathways of NQPF on RGTI. Unlike traditional productivity research, this paper focuses on the emerging concept of NQPF, deepening the academic understanding of its role in driving green innovation and contributing new insights to the theoretical framework of sustainable development and innovation. Second, by introducing IPP as a mediating variable, this study reveals the mechanism by which NQPF indirectly promote RGTI through the enhancement of IPP. The spatial mediation model presented in this paper offers a fresh perspective to existing research, particularly by providing new theoretical support for explaining how institutional safeguards can enhance regional innovation capabilities. Furthermore, as there is currently no unified approach to measuring the development level of NQPF, this study aims to establish a provincial-level index system for digital economic development from the perspectives of technological productivity, green productivity, and digital productivity. Lastly, based on panel data from 31 Chinese provinces from 2011 to 2022, this paper constructs a SDM and a spatial mediation model to examine the regional innovation spillover effects of NQPF. The empirical results not only confirm the spatial spillover effects of NQPF on green technological innovation but also provide strong policy implications for promoting interregional collaboration and the rational allocation of innovation resources, offering high practical significance.

## 2. Literature review and theoretical hypotheses

The intensification of global climate change and resource shortages has made green technological innovation an increasingly important driver of sustainable regional economic development. Green technological innovation not only effectively reduces environmental pollution and carbon emissions but also enhances resource utilization efficiency, facilitating the green transformation and upgrading of regional industrial structures [13]. In this context, how to promote green technological innovation through productivity improvement has become a focal point for both academia and policymakers [14,15]. NQPF, as a novel form of productivity, represent a transformation centered on technological innovation, digital transformation, and resource optimization [16]. Compared to traditional productivity, NQPF prioritize not only economic growth but also environmental protection and the sustainable use of resources, making them inherently aligned with green technological innovation [16]. By improving resource utilization efficiency, reducing energy consumption, and fostering technological progress, NQPF provide strong technical support for RGTI [17].

Existing literature generally agrees that NQPF can significantly promote multiple dimensions of green technological innovation. On one hand, NQPF, by advancing information technologies such as big data and the Internet of Things (IoT) and fostering intelligent manufacturing, significantly enhance the greening and intelligence of production processes. This, in turn, reduces the research and development costs of green technological innovation and improves innovation efficiency [18–20]. On the other hand, supported by digitalization and technological innovation, NQPF not only drive innovation within individual regions but also promote cross-regional cooperation and technology dissemination through technological diffusion and innovation spillovers [21–23]. These spillover effects help boost green technological innovation across regions, facilitating coordinated regional development. For example, Zhao et al. (2024) and Wicki et al. (2019) found that NQPF, through intelligent manufacturing technologies and digital transformation, enhanced corporate innovation efficiency [24,25], significantly reduced research and development costs, and minimized resource consumption, thereby driving innovation. Although existing studies have revealed

the potential relationship between NQPF and green technological innovation, there remains a lack of systematic analysis on the specific mechanisms and cross-regional spillover effects involved.

Based on the above literature, this study proposes the following hypothesis:

Hypothesis H1: NQPF have a direct impact on the development of RGTI.

In the process of promoting green technological innovation, IPP plays an indispensable role as an important institutional safeguard [26]. Firstly, new quality productive forces (NQPF) accelerate technological innovation and industrial upgrading, introducing more complex demands for intellectual property protection. This drives the expansion of protection coverage and the refinement of institutional frameworks [27]. For instance, with the advancement of green and digital technologies, emerging innovations require more comprehensive protection mechanisms to address the challenges posed by rapid technological iteration and cross-sector integration [28]. Secondly, the enhancement of NQPF intensifies inter-regional technological competition, increasing the reliance on robust intellectual property protection to ensure the equitable distribution of innovation outcomes and the controlled diffusion of technologies [29]. Furthermore, the diversified characteristics of NQPF, such as digitalization and greening, impose reform demands on traditional intellectual property protection systems, driving continuous improvements in legal frameworks and enforcement mechanisms [30]. Therefore, NQPF serves not only as a critical driving force for intellectual property protection reform but also as a fundamental pillar in enhancing regional and national competitiveness.

IPP provides legal guarantees for innovation outcomes, incentivizing firms to increase their investments in innovation. This is particularly crucial in technology- and capital-intensive industries, where IPP ensures that innovators receive the appropriate economic returns, reducing the risks associated with innovation and further enhancing their motivation [31–33]. When NQPF drive technological innovation, IPP offers robust institutional support by effectively safeguarding innovation outcomes, ensuring that these outcomes can be successfully transformed and disseminated. Research shows that strong IPP significantly promotes green technological innovation, especially in high-tech industries, by boosting innovators' confidence and expected returns, thus encouraging sustained investment in green technology innovation [34–36]. IPP also prevents excessive diffusion of innovation outcomes, ensuring that innovators reap the benefits of their patents, which further stimulates corporate innovation [37,38]. Additionally, IPP enhances the efficiency of green technology commercialization, facilitating the application and promotion of innovation outcomes, which is crucial for advancing green technological innovation [39–41].

Moreover, IPP can enhance the impact of NQPF by increasing the market value of innovation outcomes and improving the efficiency of technology commercialization in the process of promoting RGTI. The technological advancements and resource optimization driven by NQPF, when safeguarded by IPP mechanisms, can effectively amplify the spillover effects of innovation, fostering regional technological diffusion and collaboration [42]. Thus, IPP not only serves as an essential institutional safeguard for green technological innovation but also plays a critical mediating role in the process where NQPF drive green innovation. For instance, Xu et al. (2020) found through empirical research that in regions with robust IPP mechanisms, corporate investments in green technological innovation significantly increased, and the commercialization efficiency of innovation outcomes was enhanced [43]. Nguyen et al. (2023) further pointed out that IPP reduces risks associated with technology diffusion by providing legal safeguards, ensuring that the impact of NQPF on green technological

innovation is fully realized [44]. Therefore, IPP serves as an important mediator between NQPF and green technological innovation, directly promoting green innovation while also indirectly enhancing the innovation effects of NQPF by improving the efficiency of commercialization and the scope of diffusion.

Based on this, the following hypothesis is proposed:

Hypothesis H2: IPP mediates the relationship between NQPF and RGTI.

In summary, existing research has demonstrated the crucial role of NQPF in promoting green technological innovation. By improving resource utilization efficiency and driving the green transformation of production methods, NQPF have a significant positive impact on RGTI. However, discussions on the mechanisms and spillover effects of NQPF across different regions remain insufficient, especially regarding how IPP mechanisms can enhance their role in promoting green technological innovation, which has yet to receive adequate empirical support. Furthermore, there is limited research on the mediating role of IPP as an institutional safeguard in the relationship between NQPF and green technological innovation. From this perspective, the present study aims to further explore the mediating effect of IPP in the relationship between NQPF and RGTI. Through empirical analysis, this study seeks to reveal the pathways of this mechanism, as depicted in Fig 1.

## 3. Research design

This study employs the most suitable spatial regression model and a multi-period Difference-in-Differences (DID) model to analyze the panel data, in order to examine the relationships between various factors. The research methodology is illustrated in the flowchart shown in Fig 2.

### 3.1. Variable description

#### 3.1.1. Dependent variable: RGTI.
The dependent variable in this study is RGTI. RGTI refers to innovative activities that promote the efficient utilization of resources, pollution reduction and ecological protection through the research and development, application and promotion of green technologies in a specific region with the goal of sustainable development

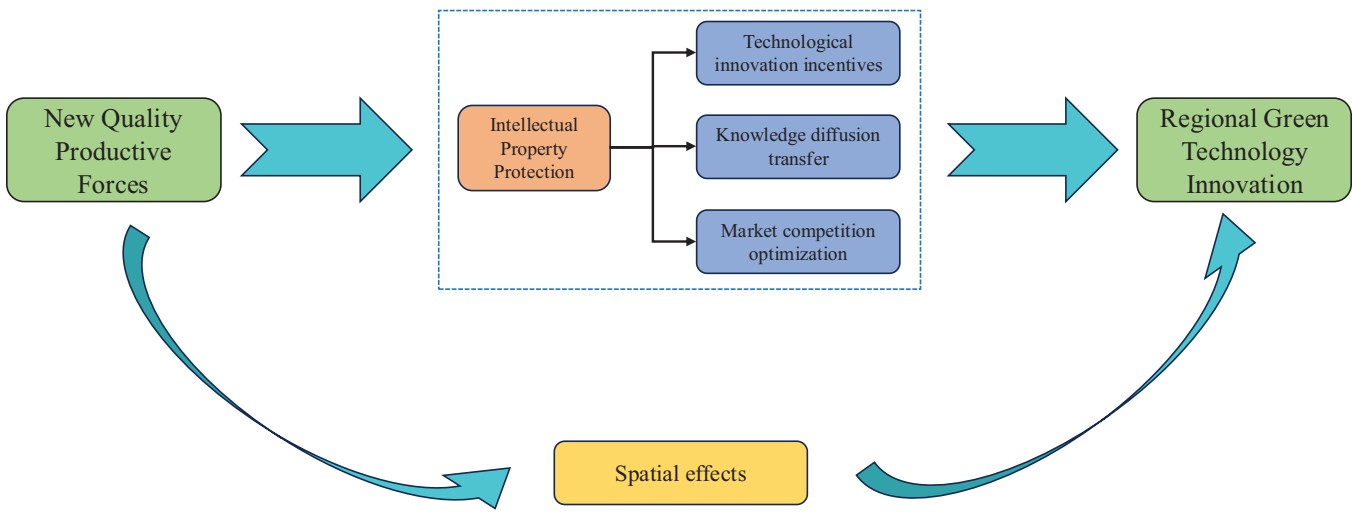

**Fig 1. Theoretical mechanism diagram.**

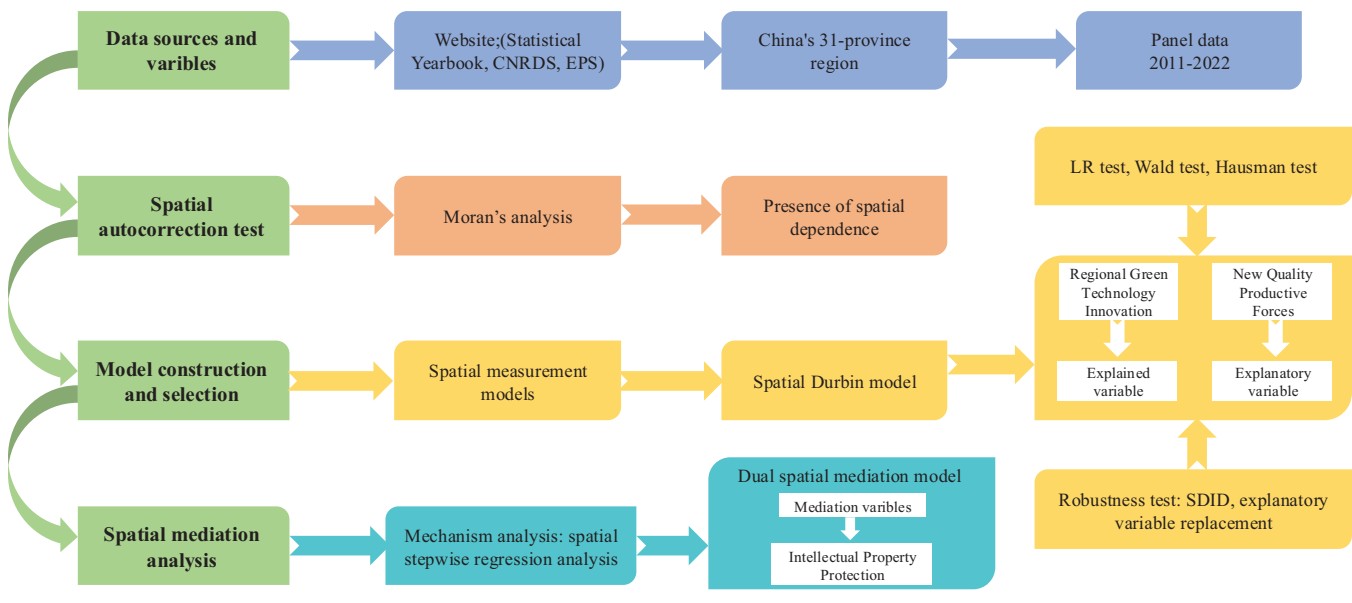

**Fig 2. Research methodology framework diagram.**

[45]. Existing research often uses the number of green patent applications [46–48] and the number of granted green patents [49] as indicators for measuring RGTI. More specifically, the China National Intellectual Property Administration classifies green patents into three categories: invention patents, utility model patents, and design patents. Compared to utility models and design patents, invention patents possess higher levels of innovation and practicality, making them more suitable for evaluating RGTI. Based on the homogeneity analysis of patent applications by Song et al. (2021) [50], this study measures the quality of regional green innovation using the number of green invention patent applications per 10,000 people in each province of China. Additionally, the quantity of regional green innovation is measured by the number of green patent applications per 10,000 people in each province. The relevant data can be obtained by consulting the IPC Green Inventory published by WIPO, following the classification numbers listed in the inventory.

**3.1.2. Independent variable: NQPF.** The independent variable in this study is NQPF. NQPF refers to a new form of productivity in the modern economic system with knowledge, technology, green development and digitalization as the core drivers [27]. Currently, there is no unified standard or method for measuring the development level of NQPF [51,52]. This study focuses on the relative level of development of NQPF at the macro level, rather than its absolute scale. From the three dimensions of technological productivity, green productivity, and digital productivity, this study constructs a three-tier system with 18 variables. The specific indicators at each level are presented in Table 1. Following the method of Xu et al. (2022) [53], this study uses an entropy-weighted TOPSIS model to calculate the specific values of NQPF for the 31 provinces of China based on collected data (detailed in Appendix A).

**3.1.3. Control variables.** To account for the various factors influencing regional green technology innovation (RGTI), this study incorporates several control variables into the model to ensure the robustness and reliability of the empirical analysis. First, the level of economic development, measured as the logarithm of per capita GDP, reflects a region's financial and infrastructural capacity to support green innovation. Second, the urbanization rate, defined as the proportion of the urban population to the total

**Table 1. NQPF development level index system.**

| Primary indicator | Secondary indicator | No. | Tertiary indicator | Unit |
|---|---|---|---|---|
| New Quality Productive Forces | Technological Productivity | A1 | Number of domestic patent authorizations | units |
| | | A2 | Revenue from high-tech industries | 10,000 yuan |
| | | A3 | Industrial innovation operations of above-scale industrial enterprises | 10,000 yuan |
| | | A4 | Labor productivity of above-scale industrial enterprises | % |
| | | A5 | Number of above-scale industrial enterprises engaged in R&D activities | units |
| | | A6 | Original density of machine installations | % |
| | Green Productivity | B1 | Energy consumption per unit of GDP | % |
| | | B2 | Fossil energy consumption per unit of GDP | % |
| | | B3 | Industrial water use per unit of GDP | % |
| | | B4 | Comprehensive utilization of industrial solid waste per unit of production | % |
| | | B5 | Industrial wastewater discharge per unit of GDP | % |
| | | B6 | Industrial SO2 emissions per unit of GDP | % |
| | Digital Productivity | C1 | Integrated circuit production | 10,000 yuan |
| | | C2 | Total telecommunications business volume | 10,000 yuan |
| | | C3 | Number of internet broadband access users | 10,000 households |
| | | C4 | Software business revenue | 10,000 yuan |
| | | C5 | Length of fiber optic cable lines | kilometers |
| | | C6 | E-commerce sales | 10,000 yuan |

population, captures agglomeration effects that attract talent and resources, thereby enhancing regional innovation capacity. Third, the industrial structure, represented by the proportion of the tertiary sector in regional GDP, indicates industrial advancement, which can facilitate technological innovation and diffusion. Fourth, the energy consumption structure, measured as the proportion of coal consumption in total energy consumption, reflects the region's reliance on clean energy and its application of green technologies. Fifth, technological investment, calculated as the proportion of local government fiscal expenditure on science and technology, represents government support for innovation, which directly affects green innovation outcomes. Sixth, the intensity of environmental regulation, proxied by the proportion of industrial pollution control investments in industrial added value, incentivizes firms to adopt green technologies and improve innovation efficiency. Finally, human capital, expressed as the number of full-time teachers in higher education institutions per 10,000 people, captures the region's technical expertise and innovation capacity, both critical for driving green innovation. These variables, sourced from national and regional statistical yearbooks, provide a comprehensive framework to control for the confounding effects of regional heterogeneity in the analysis.

**3.1.4. Mediating variable.** IPP refers to a series of systems and measures to provide legal protection for intellectual achievements and identifying resources created by individuals or organizations in the scientific, technological, cultural and commercial fields through legal, administrative and technical means [54]. To clarify the role of IPP in the relationship between NQPF and green technological innovation, this study incorporates IPP as a mediating variable. The data for IPP are sourced from the IPP index system and test indices for 31 Chinese provinces, as constructed in the "Intellectual Property Development Status Evaluation Report" published by the Intellectual Property Development Research Center of the China National Intellectual Property Administration. The specific variable descriptions are provided in Table 2.

**Table 2. Variable descriptions.**

| Classification | Declaration | Symbol | Definitions |
|---|---|---|---|
| Dependent variable | Regional green technology innovation | rgti | Number of green invention patent applications per 10,000 people. |
| Independent variable | New quality productive forces | nqpf | Calculated based on the entropy-weighted TOPSIS method. |
| Control variables | Economic development level | lnpgdp | Logarithm of per capita GDP. |
| | Urbanization rate | ur | Percentage of urban population (%) |
| | Industrial structure | inds | Proportion of the tertiary sector in GDP. |
| | Energy consumption structure | ecs | Proportion of coal consumption in total energy consumption. |
| | Technological investment | teci | Proportion of local government fiscal expenditure on science and technology in total fiscal expenditure. |
| | Environmental regulation intensity | eri | Proportion of industrial pollution control investment in industrial added value. |
| | Human capital | hc | Number of full-time teachers in higher education institutions (10,000 people). |
| Mediating variable | Intellectual property protection | ipp | Intellectual property protection index. |

## 3.2. Data sources

Considering the rapid development of NQPF and green technological innovation over the past decade, as well as the availability of relevant provincial-level data, this study selects panel data from 31 provinces between 2011 and 2022 as the research sample. The data primarily come from the China Regional Statistical Yearbook, Provincial Statistical Yearbooks, the CNRDS (China Research Data Service Platform), the Intellectual Property Development Status Evaluation Report, and the EPS Global Data Statistics Platform. For a few missing data points from certain regions in Tibet, we employed interpolation and median methods to fill in the gaps. To analyze the data, we used STATA 17.

## 3.3. Model selection and construction

The theoretical analysis above suggests that there may be spatial correlations between NQPF and RGTI. Ignoring spatial effects could lead to biased empirical results. However, it is difficult to directly determine whether the digital economy promotes or inhibits green innovation in local and surrounding areas. Therefore, it is necessary to construct an appropriate spatial econometric model to explain this relationship. Compared to other spatial econometric models, the SDM is a standard structure that can capture various types of spatial spillover effects. Under different coefficient settings, it can be transformed into the commonly used Spatial Lag Model (SLM) and Spatial Error Model (SEM), making it more suitable for general use. Consequently, this study constructs the following SDM for empirical testing:

**3.3.1. Spatial durbin model.** To investigate the direct and indirect effects of NQPF on RGTI across 31 Chinese provinces, we construct the following spatial econometric model:

$$rgti_{it} = \alpha + \rho \sum_{j=1}^{n} W_{ij}^{de} rgti_{jt} + \beta_0 nqpf_{it} + \beta_1 Y_{it} + \mu_i + \varphi_i + \varepsilon_{it} \tag{1}$$

The SLM, SEM, and SDM are the three primary components of spatial econometric models. Compared to other spatial econometric models, the SDM is a standard structure capable of capturing various types of spatial spillover effects. Depending on different coefficient settings, it can be transformed into the commonly used SLM and SEM, making it more suitable for general use. Therefore, this study develops a SDM with both time and spatial effects for empirical testing:

$$rgti_{it} = \alpha + \rho\sum_{j=1}^{n}W_{ij}^{de}rgti_{jt} + \beta_0 nqpf_{it} + \beta_1 Y_{it} + \theta_0\sum_{j=1}^{n}W_{ij}^{de}nqpf_{it} + \theta_1\sum_{j=1}^{n}W_{ij}^{de}Y_{it} + \mu_i + \varphi_i + \varepsilon_{it}$$

(2)

In Model (2), $i$ denotes the year, $t$ represents the province, $\alpha$ is the constant term, $\rho$ and $\theta_i$ represent the spatial lag regression coefficients, $\beta_i$ represents the regression coefficients, $\mu_i$ and $\varphi_i$ represent the spatial and temporal fixed effects, and $\varepsilon_{it}$ denotes the random disturbance term. Here, $rgti$ refers to RGI, $nqpf$ represents NQPF, $Y$ to the control variables, and $W_{ij}^{de}$ to the geographically and economically nested spatial weight matrix.

This study will use LR and Wald tests to verify the appropriateness of model selection. If the results reject the null hypothesis, it indicates that the SDM does not degenerate into SLM or SEM, thus supporting the choice of the model in this study. The Hausman test will then be applied to determine whether fixed effects or random effects are more appropriate. Finally, the study employs the maximum likelihood estimation method to ensure the accuracy of the results.

**3.3.2. Spatial mediation model.** This study employs a non-linear mediation model to examine how carbon IPP influences the relationship between NQPF and RGTI. Therefore, the spatial mediation model constructed in this study is used for empirical testing:

$$M_{it} = \alpha + \beta_0 nqpf_{it} + \beta_1 Y_{it} + \theta_0\sum_{j=1}^{n}W_{ij}^{de}nqpf_{it} + \theta_1\sum_{j=1}^{n}W_{ij}^{de}Y_{it} + \mu_i + \varphi_t + \varepsilon_{it}$$

(3)

$$rgti_{it} = \alpha + \rho\sum_{j=1}^{n}W_{ij}^{de}rgti_{it} + \beta_0 nqpf_{it} + \beta_1 Y_{it} + \delta M_{it} + \theta_0\sum_{j=1}^{n}W_{ij}^{de}nqpf_{it} + \theta_1\sum_{j=1}^{n}W_{ij}^{de}Y_{it} + \gamma\sum_{j=1}^{n}W_{ij}^{de}M_{it} + \mu_i + \varphi_i + \varepsilon_{it}$$

(4)

In model (4), $M$ represents the mediating variable.

**3.3.3. Multi-period SDID model.** To verify the robustness of the results from the SDM and the Spatial Mediation Model, this study replaces the independent variable with the "National Big Data Experimental Zone Construction" pilot policy. This policy serves as an exogenous shock to assess the impact of NQPF on RGTI. Since the pilot policy was implemented in two phases, a multi-period SDID model is constructed to analyze the spatial effects of the pilot policy on RGTI.

$$rgi_{it} = \alpha + \rho\sum_{j=1}^{n}W_{ij}^{de}rgti_{jt} + \beta_0\left(bigdata_i \times post_t\right) + \beta_1 Y_{it} + \theta_0\sum_{j=1}^{n}W_{ij}^{de}\left(bigdata \times post\right)_{it} + \theta_1\sum_{j=1}^{n}W_{ij}^{de}Y_{it} + \mu_i + \varphi_i + \varepsilon_{it}$$

(5)

In Model (5), $bigdata_i$ is a dummy variable indicating whether the region is part of the pilot policy; it takes the value "1" if the region is included in the policy and "0" otherwise. $post_t$ is a dummy variable representing the time of policy implementation; it takes the value "1" for years following the implementation of the policy, and "0" otherwise.

## 4. Results and analysis

### 4.1. Descriptive statistics

We first conducted descriptive statistical analysis on all variables to gain an intuitive understanding of the data distribution and characteristics, and to assess data quality. The results of the descriptive statistical analysis are presented in Table 3. The findings indicate that no data set exhibits abnormally high or low values, the degree of data dispersion is low, and the data distribution trends are favorable.

**Table 3. Descriptive statistics of variables.**

| Variable | Obs | Mean | Std.Dev. | Min | Max |
|---|---|---|---|---|---|
| rgti | 372 | 1.611 | 1.581 | 0.278 | 10.763 |
| nqpf | 372 | 0.196 | 0.179 | 0.027 | 0.877 |
| nqpf$^2$ | 372 | 0.07 | 0.137 | 0.001 | 0.769 |
| ipp | 372 | 65.833 | 14.03 | 0.406 | 93.74 |
| lnpgdp | 372 | 10.857 | 0.463 | 9.682 | 12.156 |
| ur | 372 | 0.592 | 0.132 | 0.228 | 0.938 |
| inds | 372 | 1.362 | 0.735 | 0.527 | 5.283 |
| ecs | 372 | 0.957 | 0.584 | 0.012 | 6.211 |
| teci | 372 | 0.021 | 0.015 | 0.003 | 0.068 |
| eri | 372 | 2.66 | 2.788 | 0.076 | 14.638 |
| hc | 372 | 0.003 | 0.004 | 0.001 | 0.031 |

To illustrate the spatial effects of NQPF on RGTI, we conducted spatial autocorrelation analysis for each of these two variables and then identified the spillover effects at the city level through spatial variation trends. The entropy-weighted TOPSIS model was used to measure the indicator system of nqpf development levels.

Tobler's First Law of Geography states that "everything is related to everything else, but near things are more related than distant things" [55]. Therefore, the most commonly used methods to explore spatial correlation are the adjacency weight matrix, the inverse distance weight matrix, and the economic distance weight matrix. There is currently no consensus on which spatial weight matrix is optimal [56]. Considering that NQPF may be influenced by both geographic and economic factors in their impact on RGTI, this study follows the approach of Case et al. (1993) [57], constructing an inverse distance spatial weight matrix and an economic distance spatial weight matrix, and then nesting the two to form a geographic-economic nested weight matrix. The specific expression is as follows:

$$W_{ij}^{de} = \begin{cases} \alpha \times W_{ij}^d + (1-\alpha) \times W_{ij}^e & i \neq j \\ 0 & i = j \end{cases} \quad (6)$$

In [Equation (6)], $W_{ij}^d = 1/d_{ij}$ represents the inverse distance spatial weight matrix, indicating that as the geographical distance between two regions decreases, their correlation becomes stronger. $W_{ij}^e = 1/|pgdp_i - pgdp_j|$ represents the economic distance spatial weight matrix, indicating that the correlation is stronger when the per capita GDP of the two regions is more similar.

Based on the constructed weight matrix, we used Moran's I index to test the spatial auto-correlation of regional green innovation. The global Moran's I value ranges between [-1, 1], where a positive value indicates positive spatial correlation, and a negative value indicates negative spatial correlation. According to the results in [Table 4], the Moran's I index for regional green innovation and NQPF, using the geographic-economic nested weight matrix, successfully passed the 1% significance test. This indicates a significant spatial clustering characteristic of regional green innovation across the provinces.

## 4.2. Spatial impact of NQPF on RGTI

OLS regression allows for distinguishing the roles of different variables in influencing RGTI, and the coefficients can be interpreted as drivers of regional green innovation. However, OLS

Table 4. Global Moran's I index for RGTI and NQPF.

| year | RGTI | | NQPF | |
|---|---|---|---|---|
| | Moran's I | Z value | Moran's I | Z value |
| 2011 | 0157*** | 5.321 | 0.085*** | 2.532 |
| 2012 | 0.164*** | 5.272 | 0.09*** | 2.634 |
| 2013 | 0.148*** | 5.268 | 0.094*** | 2.699 |
| 2014 | 0.124*** | 4.986 | 0.105*** | 2.925 |
| 2015 | 0.155*** | 5.338 | 0.111*** | 3.080 |
| 2016 | 0.171*** | 5.385 | 0.114*** | 3.185 |
| 2017 | 0.15*** | 4.941 | 0.099*** | 2.877 |
| 2018 | 0.144*** | 4.938 | 0.09*** | 2.726 |
| 2019 | 0.144*** | 5.207 | 0.095*** | 2.843 |
| 2020 | 0.139*** | 5.049 | 0.098*** | 2.858 |
| 2021 | 0.123*** | 4.911 | 0.099*** | 2.886 |
| 2022 | 0.107*** | 4.731 | 0.094*** | 2.772 |

*, **, and *** indicate significance at the levels of 10, 5 and 1%, respectively.

regression overlooks the distance effect. In column (1), the estimated coefficients for both the primary and secondary terms of the core explanatory variable are significantly larger than those in column (2), indicating that the OLS method fails to capture the spatial impact of NQPF on RGTI. Since the estimated results exhibit bias, we then apply the SLM, SEM, and SDM to demonstrate the spatial autocorrelation of green innovation. Based on previous model selection, we conducted LR and Wald tests, both of which rejected the initial hypotheses, indicating that using SDM is preferable to SLM and SEM, and that the model does not degenerate. The Hausman test results further ruled out the use of fixed or random effects models [58], as the Hausman test results are applicable not only to linear models but also to the maximum likelihood method. As shown in Table 5, the Hausman test results indicate an equivalent statistic of 18 (p = 0.035), and the null hypothesis is rejected at the 5% significance level, leading to the final selection of the SDM fixed effects model. Given the spatial correlation, this study uses the maximum likelihood method to estimate the spatial impact of nqpf on RGTI based on model selection, and the spatial effects are analyzed in terms of direct, indirect, and overall impacts.

The regression results with fixed spatial and temporal effects are presented in Table 6. Due to the regional spillover effects among China's 31 provinces, NQPF and IPP cannot be regarded as independent factors influencing regional green innovation. The estimation results show that the spatial rho is negatively correlated and statistically significant at the 5% significance level. Furthermore, in column (2), the main effect of the SDM shows that the coefficient for nqpf is 1.752, which is statistically significant at the 10% significance level. This positive effect reflects how higher levels of productive forces provide firms with more technological innovation resources and capital accumulation, thereby fostering regional innovation activities. Moreover, the coefficient for $nqpf^2$ is -4.847, statistically significant at the 1% level, indicating a significant negative effect of the squared level of nqpf on innovation. This suggests that while the initial development of nqpf promotes innovation, excessive development leads to a decline in innovation efficiency once it reaches a certain level. This decline could be attributed to resource constraints, rising management costs, innovation path dependency, or changes in external economic conditions.

However, the Wx effect of NQPF in Column (4) is not significant (1.857, p > 0.1), indicating that the direct spatial spillover effect is limited. The quadratic term of NQPF in the Wx effect

**Table 5. Spatial effects of the NQPF on RGTI.**

| Variables | OLS | SDM | | | | |
|---|---|---|---|---|---|---|
| | | Main | Wx | Direct effect | Indirect effect | Total effect |
| | (1) | (2) | (3) | (4) | (5) | (6) |
| nqpf | 2.999*** | 1.752* | 1.857 | 1.735* | 0.454 | 2.189 |
| | (0.587) | (1.003) | (7.776) | (0.935) | (5.586) | (5.955) |
| nqpf² | -2.06*** | -4.847*** | -23.511*** | -4.434*** | -16.331*** | -20.765*** |
| | (0.633) | (1.09) | (6.823) | (0.951) | (4.855) | (5.119) |
| lnpgdp | 0.17* | 1.763*** | 2.085 | 1.755*** | 1.04 | 2.795 |
| | (0.091) | (0.388) | (2.129) | (0.399) | (1.813) | (1.734) |
| ur | 1.505*** | -1.995** | -56.1*** | -0.686 | -42.499*** | -43.185*** |
| | (0.279) | (0.93) | (5.741) | (0.96) | (5.069) | (5.148) |
| inds | 0.987*** | 0.302*** | -1.366* | 0.315** | -1.117** | -0.802 |
| | (0.126) | (0.118) | (0.765) | (0.131) | (0.554) | (0.553) |
| ecs | 0.021 | -0.032 | -0.823 | -0.014 | -0.637 | -0.65 |
| | (0.028) | (0.069) | (0.525) | (0.066) | (0.437) | (0.447) |
| teci | 19.006*** | 21.989*** | 15.872 | 21.982*** | 6.705 | 28.687** |
| | (3.321) | (3.672) | (16.963) | (3.535) | (13.225) | (13.282) |
| eri | -0.097*** | -0.332*** | -0.774*** | -0.326*** | -0.487*** | -0.813*** |
| | (0.026) | (0.034) | (0.177) | (0.03) | (0.131) | (0.135) |
| hc | 8.311 | 9.009 | 16.809 | 8.867 | 11.495 | 20.362 |
| | (5.908) | (6.697) | (31.905) | (5.918) | (23.421) | (22.795) |
| Spatial rho | | 0.343** | | | | |
| | | (0.159) | | | | |
| Wald_lag | | 120.00 (Prob> chi2 = 0.000) | | | | |
| Wald_err | | 122.44 (Prob> chi2 = 0.000) | | | | |
| LR_lag | | 96.51 (Prob> chi2 = 0.000) | | | | |
| LR_err | | 100.88 (Prob> chi2 = 0.000) | | | | |
| Hausman | | 18.00 (Prob> chi2 = 0.035) | | | | |
| Spatial FE | Yes | Yes | Yes | Yes | Yes | Yes |
| Time FE | Yes | Yes | Yes | Yes | Yes | Yes |
| Observations | 372 | 372 | 372 | 372 | 372 | 372 |
| R-squared | 0.797 | 0.431 | 0.431 | 0.431 | 0.431 | 0.431 |
| Number | 31 | 31 | 31 | 31 | 31 | 31 |

Values in parentheses represent standard errors; for instance, the number within (0.587) indicates the standard error of the corresponding coefficient. *, **, and *** indicate significance at the levels of 10, 5 and 1%, respectively.

is significantly negative (-23.511, $p < 0.01$), suggesting that high levels of NQPF in neighboring regions may inhibit local innovation. This could potentially result from resource competition or path dependency in innovation activities. As indicated by the significant direct effects in Column (4), the inverted "U" relationship of NQPF is primarily driven by local dynamics. Although the spatial lag effect of NQPF itself is not significant, the significantly negative quadratic term suggests that high levels of NQPF in neighboring regions may hinder local green innovation. This phenomenon may stem from inter-regional competition for innovation resources or a crowding-out effect caused by the excessive concentration of high-level productive forces.

In columns (5) and (6), the significance of NQPF decreases or disappears, indicating that when considering the spillover effects from neighboring regions, the direct promoting effect

**Table 6. Regression results of the spatial mediation model.**

| Model | Model 1 | Model 2 | Model 3 | Model 4 | Model 5 |
|---|---|---|---|---|---|
| Variables | rgti | rgti | ipp | ipp | rgti |
| nqpf | 0.460 | 1.735* | 20.396* | 27.866** | 1.569 |
| | (1.597) | (0.935) | (11.15) | (13.14) | (6.422) |
| nqpf² | 3.662** | -4.433*** | -57.855*** | -62.435*** | -16.049*** |
| | (1.593) | (0.951) | (11.723) | (13.185) | (5.553) |
| ipp | | | | | 0.046*** |
| | | | | | (0.016) |
| lnpgdp | | 1.755*** | | -12.433** | 1.562 |
| | | (0.399) | | (5.078) | (1.728) |
| ur | | -0.686 | | 4.96 | -46.477*** |
| | | (0.96) | | (13.272) | (6.381) |
| inds | | 0.315** | | -2.291 | -0.982* |
| | | (0.131) | | (1.743) | (0.581) |
| ecs | | -0.014 | | 1.877** | -0.8* |
| | | (0.066) | | (0.859) | (0.411) |
| teci | | 21.982*** | | -100.115** | 12.368 |
| | | (3.535) | | (46.696) | (12.898) |
| eri | | -0.326*** | | -1.381*** | -0.45*** |
| | | (0.03) | | (0.393) | (0.131) |
| hc | | 8.867 | | -295.394*** | 25.52 |
| | | (5.918) | | (77.750) | (27.573) |

Values in parentheses represent standard errors; for instance, the number within (1.597) indicates the standard error of the corresponding coefficient. *, **, and *** indicate significance at the levels of 10, 5 and 1%, respectively.

of nqpf development on green innovation may be weakened or offset. However, the squared term shows a significant negative effect at the 1% level, exhibiting non-linear characteristics. This suggests that even though the direct promoting effect of nqpf on green technological innovation may not be significant, as the level of nqpf increases, its diminishing effect is significantly amplified through spatial effects. This negative impact may be due to the saturation of innovation, caused by the over-allocation of resources or the over-exploitation of innovation capacity.

## 4.3. Mediation effect analysis

We employed a stepwise regression model with time and spatial fixed effects to determine how GDP influences eco-innovation through the mediating effects of carbon emissions and FDI. The results of the hypotheses proposed in this study are presented in Table 6.

Models 1 and 2 examine the impact of NQPF on RGTI. In Model 1, nqpf is not statistically significant, meaning it cannot explain the relationship through statistical significance, whereas its squared term is significant at the 5% level, with a coefficient of 3.662. This indicates a non-linear relationship: at lower levels of nqpf, its increase may not have a significant effect, but at certain thresholds, the positive effect of the squared term suggests that as nqpf continues to grow, it may start to have a stronger impact on green technological innovation. Model 2 shows that after adding some control variables, nqpf becomes significant at the 10% level, with a coefficient of 1.735, indicating that after accounting for control variables, nqpf has a significant positive effect on RGTI. The squared term also becomes more significant, though with a

coefficient of -4.433, suggesting that as nqpf continues to rise, its positive effect on innovation starts to diminish and may even turn negative. This means that the continuous growth of nqpf could ultimately lead to a decline in innovation activity due to factors like resource constraints or diminishing efficiency.

Models 3, 4, and 5 assess the impact of NQPF on IPP and the mediating role of IPP in the relationship between nqpf and RGTI. Model 3 shows that, without control variables, there is a significant positive correlation between nqpf and IPP at the 5% significance level (20.396), indicating that an increase in nqpf contributes to strengthening IPP. However, the squared term of nqpf exhibits a significant negative correlation at the 1% level (-57.855), reflecting a non-linear characteristic. This may suggest that when nqpf reaches a higher level, issues such as technological blockades or uneven resource allocation may arise, thereby weakening the effectiveness of IPP. Model 4 shows that, with the inclusion of control variables, the positive correlation between nqpf and IPP remains significant at the 5% level (27.866), and the squared term remains significantly negative at the 1% level. The regression results are consistent with Model 3, indicating that after adding control variables, the negative impact of nqpf on IPP becomes more pronounced once nqpf reaches a certain level. In Model 5, after introducing the mediating variable IPP, the direct relationship between nqpf and RGTI becomes insignificant, suggesting that the direct effect of nqpf is weakened or disappears. This may be due to IPP acting as the primary mediator in this model. The coefficient of IPP is 0.046 and is significantly positive at the 1% level, indicating that IPP has a significant positive impact on RGTI. In other words, nqpf indirectly affects RGTI through IPP, reinforcing the crucial role of IPP as a driver of innovation. However, the squared term of nqpf remains significantly negative at the 1% level, suggesting that under the influence of the mediating variable IPP, the direct effect of nqpf is weakened or offset, and the excessive concentration of innovation resources or increased complexity in management leads to negative spillover effects.

This study, through analyzing the effects of NQPF and IPP on RGTI, reveals that the development of nqpf initially has a significant positive impact on green technological innovation. However, as the level of productivity reaches a certain threshold, its positive effect gradually diminishes and even turns negative, demonstrating a notable inverted U-shaped, non-linear pattern. Furthermore, IPP, as a mediating factor, significantly enhances the indirect effect of nqpf on green technological innovation, providing empirical support for the development of more effective innovation protection mechanisms. Finally, the scope of this study extends beyond the relationship between nqpf and RGTI by integrating the interactive effects of IPP and innovation, offering valuable policy insights for future regional innovation development. In particular, the study provides insightful recommendations on how to balance productivity growth with the rational allocation of innovation resources, contributing valuable perspectives for further research.

### 4.4. Robustness test

**4.4.1. Exogenous shock test.** To verify the robustness of the results mentioned above, this study replaces the independent variable with the "National Big Data Pilot Zone Construction" policy as an external shock to evaluate the impact of NQPF on RGTI. In February 2016, the Chinese government approved Guizhou Province to establish the first national-level comprehensive big data pilot zone; in October of the same year, the second batch of national big data pilot zones was launched in seven regions, including Beijing-Tianjin-Hebei, Inner Mongolia, and Liaoning. These eight national big data pilot zones, spanning from south to north and east to west, formed the "three-dimensional framework" of China's big data development practices. This has played a key role in accelerating the innovation landscape

driven by nqpf and enhancing RGTI. Since the pilot policy was implemented in two phases, this study applies the multi-period SDID model constructed earlier to analyze the spatial impact of the pilot policy on regional green innovation.

The exogenous shock test focuses solely on whether the impact of NQPF on regional green innovation is promotive or inhibitive, and whether spatial spillover effects exist. As shown in Table 7, the implementation of the "National Big Data Pilot Zone Construction" policy improves RGTI under the geographic-economic nested spatial weight matrix, consistent with the results of the SDM regression. This indicates that the development of nqpf has a spillover effect, enhancing both the geographic and economic impacts on RGTI. The results also meet the conditions of the parallel trend and placebo checks, confirming the reliability of this robustness test.

**4.4.2. Replacement of the dependent variable.** In the baseline model, we used the number of green invention patent applications per 10,000 people to measure RGTI, representing the "quality" of regional green innovation. In fact, most researchers also use the "quantity" of RGTI to measure regional green innovation, which includes the total number of invention, utility model, and design patent applications. Therefore, to avoid the influence of different indicator selections on the robustness of the results, this study replaces the initial dependent variable with the total number of green patent applications per 10,000 people. We then performed the SDM regression again. The robustness test results are shown in Table 8. The spatial rho is negatively correlated and statistically significant at the 1% significance level. In the direct effects, the primary regression coefficient of nqpf is significantly positive at the 5% level, indicating that the development of nqpf promotes RGTI. However, the squared term is significantly negative at the 1% level, presenting a notable inverted U-shape, which reflects non-linear characteristics. This suggests that after nqpf reaches a certain stage of development, factors such as resource constraints, innovation saturation, or other bottlenecks may lead to a decline in green technological innovation. The spatial effects of the SDM model are consistent with the above regression results, confirming the robustness of the baseline results.

## 5. Conclusion and policy recommendations

### 5.1. Research conclusions

This study analyzes panel data from 31 provinces in China from 2011 to 2022 to explore the impact of NQPF on RGTI, as well as the mediating role of IPP. The findings indicate that nqpf significantly promotes RGTI by improving resource utilization efficiency and greening

**Table 7. SDID model regression results.**

| Variables | Main | Wx | Direct effect | Indirect effect | Total effect |
|---|---|---|---|---|---|
| bigdata*post | 0.145 * | 1.484*** | 0.186** | 2.21*** | 2.397*** |
| | (0.083) | (0.365) | (0.085) | (0.747) | (0.783) |
| Spatial rho | 0.286** | | | | |
| | (0.126) | | | | |
| Spatial FE | Yes | Yes | Yes | Yes | Yes |
| Time FE | Yes | Yes | Yes | Yes | Yes |
| Observations | 372 | 372 | 372 | 372 | 372 |
| R-squared | 0.468 | 0.468 | 0.468 | 0.468 | 0.468 |
| Number | 31 | 31 | 31 | 31 | 31 |

Values in parentheses represent standard errors; for instance, the number within (0.083) indicates the standard error of the corresponding coefficient. *, **, and *** indicate significance at the levels of 10, 5 and 1%, respectively.

**Table 8. Robustness test.**

| Variables | Direct effect | Indirect effect | Total effect |
|---|---|---|---|
| nqpf | 2.066** | 2.552 | 4.618 |
| | (0.966) | (5.707) | (6.08) |
| nqpf$^2$ | -4.312*** | -18.974*** | -23.286*** |
| | (0.987) | (5.016) | (5.288) |
| Spatial rho | -0.353** | | |
| | (0.159) | | |
| Spatial FE | Yes | Yes | Yes |
| Time FE | Yes | Yes | Yes |
| Observations | 372 | 372 | 372 |
| R-squared | 0.846 | 0.846 | 0.846 |
| Number | 31 | 31 | 31 |

Values in parentheses represent standard errors; for instance, the number within (0.966) indicates the standard error of the corresponding coefficient. *, **, and *** indicate significance at the levels of 10, 5 and 1%, respectively.

production processes, thereby enhancing innovation capacity. However, there is an inverted U-shaped relationship between nqpf and green innovation, the inverted "U" relationship of nqpf is primarily driven by local dynamics, meaning that after nqpf reaches a certain level, its positive effect on innovation begins to weaken, potentially leading to diminishing marginal returns. Furthermore, IPP plays a critical mediating role between nqpf and green technological innovation. Strengthening IPP not only directly promotes green technological innovation but also amplifies the positive impact of nqpf on green innovation. Therefore, the synergy between nqpf and IPP is a key factor in driving RGTI.

## 5.2. Policy recommendations

Based on the findings of this study, the following policy recommendations are proposed to further drive the development of NQPF and RGTI. These suggestions aim to address both immediate and long-term needs, ensuring that the synergy between productivity growth and innovation translates into sustainable development:

**5.2.1. Tailored support for NQPF at different development stages.** The inverted U-shaped relationship between nqpf and green innovation highlights the need for stage-specific policy interventions. In the early stages of productivity growth, the government should implement policies that incentivize technological innovation and green transformation. This can be achieved through targeted subsidies, grants, and investment in digital infrastructure to accelerate the development of emerging industries like green technology, artificial intelligence, and digital manufacturing. In contrast, for regions where productivity has reached a certain threshold, policies should focus on enhancing the efficiency of resource utilization and minimizing the negative effects of overconcentration of innovation resources. This includes promoting diversified investment in R&D and reducing potential bottlenecks associated with innovation saturation.

**5.2.2. Promoting equitable distribution and sharing of innovation resources.** The spatial spillover effects revealed in this study underscore the importance of cross-regional collaboration. Policymakers should prioritize mechanisms that facilitate the equitable distribution and sharing of innovation resources across regions. This can be achieved by establishing regional innovation hubs and promoting interregional partnerships through innovation-sharing platforms and collaborative research centers. These initiatives should

focus on reducing the concentration of innovation in a few highly developed areas, while enabling less developed regions to benefit from shared knowledge, resources, and technological advancements. Such cooperation will help to mitigate the diminishing returns seen in regions with high levels of productivity concentration.

**5.2.3. Strengthening IPP for enhanced innovation.** As IPP plays a critical mediating role in amplifying the positive effects of nqpf on green innovation, it is essential to further strengthen IPP mechanisms. Governments should enhance legal frameworks to ensure robust protection of intellectual property rights, particularly in high-tech and green industries. This can be supported through the establishment of fast-track patent processes for green technologies, increasing penalties for IP infringements, and facilitating international cooperation on intellectual property enforcement. Additionally, efforts should be made to create platforms that facilitate the commercialization and transfer of intellectual property, thereby ensuring that innovative technologies are efficiently translated into marketable products that contribute to green development.

**5.2.4. Integrating green innovation into broader regional development strategies.** Given the geographic and economic spillover effects of nqpf and IPP, policymakers should integrate green innovation into broader regional development strategies. This requires embedding green technology development goals into regional economic plans, emphasizing sustainability and resilience in economic growth strategies. Governments should encourage the development of green innovation clusters, which leverage regional strengths and foster competition, while ensuring environmental sustainability. Furthermore, policy measures should focus on capacity building for local enterprises and innovation ecosystems, ensuring that they can effectively absorb and utilize green technologies.

**5.2.5. Long-term monitoring and adaptation of policies.** The dynamic nature of nqpf and its impact on green technological innovation necessitates the continuous monitoring of policy outcomes. Policymakers should establish feedback mechanisms that allow for the periodic assessment of the effectiveness of support programs, resource allocation strategies, and IPP enforcement. This would enable timely adjustments to policies in response to evolving technological trends, market demands, and environmental challenges. By maintaining flexibility and responsiveness, governments can ensure that their policies remain effective in fostering long-term sustainable innovation.

## 5.3. Limitations and future outlook

Although this study reveals the complex interactions between NQPF, IPP and RGTI, several limitations remain. First, the data used in this study are limited to 31 provinces in China. While the findings provide valuable insights for regional innovation policies in China, their applicability at the international level still needs further validation. Second, although we employed the SDM to capture spillover effects between regions, other exogenous factors that may influence green technological innovation—such as environmental policies, international trade, and market structure—were not included in the model, potentially affecting the comprehensiveness of the conclusions. Lastly, due to the relatively short time span of the panel data, it was not possible to fully analyze the long-term effects of nqpf on green technological innovation.

Future research can explore the following aspects. First, expanding the scope of the data to conduct cross-country comparative studies could verify whether the impact of nqpf and IPP on green technological innovation is consistent internationally. Second, future studies could incorporate more exogenous variables, such as environmental policies and international trade, to more comprehensively investigate the drivers of green technological innovation. Additionally, as more data becomes available, future research could further explore the long-term

dynamic impacts of nqpf and IPP on green technological innovation, providing deeper policy insights for the sustainability of regional green development.

## Appendix A

This study employs the Entropy Weight-TOPSIS approach to evaluate the comprehensive NQPF performance in each region. The methodology involves two stages: first, the entropy weight technique assigns a weighted value to each indicator based on its standardized distribution, minimizing the impact of subjective factors; second, the TOPSIS method quantitatively ranks the NQPF level across regions by calculating the relative distance between each evaluation object and ideal or worst-case scenarios. The proposed Entropy Weight-TOPSIS approach benefits from the simplicity of calculation and the reasonableness of results obtained, providing a more objective and reasonable measurement of NQPF development levels. The detailed steps for implementation are as follows:

Step 1: Data normalization and standardization.

To address potential discrepancies in the data, negative values must be recoded as positive ones. Fortunately, the NQPF development level index framework developed in this study features 12 three-tier evaluation indicators that are inherently positive in value, thereby rendering conversion unnecessary. To reconcile differences in scale, each measurement indicator $x_{ij}$ within the NQPF development level index system is standardized initially.

$$x'_{ij} = \frac{x_{ij} - \min\left(x_j\right)}{\max\left(x_j\right) - \min\left(x_j\right)} \tag{8}$$

$$x'_{ij} = \frac{\max\left(x_{ij}\right)}{mac\left(x_j\right) - \min\left(x_j\right)} \tag{9}$$

Where $x_{ij}$ represents the value of the $j$ evaluation indicator for the $i$ city, and $x'_{ij}$ represents the standardized value of the NQPF measurement indicator.

Step 2: Calculate the information entropy $E_j$ for each standardized measurement indicator $x'_{ij}$ within the NQPF development level index System:

$$E_j = -k\sum_{i=1}^{n} R_{ij}\ln(R_{ij}) \ , \ k = \frac{1}{\ln n} \ , \ R_{ij} = \frac{x_{ij}}{\sum_{i=1}^{n} x'_{ij}} \tag{10}$$

Step 3: Calculate the weight $W_j$ of each measurement indicator in the NQPF development level index system:

$$W_j = \frac{1 - E_j}{\sum_{j=1}^{m}\left(1 - E_j\right)} \ , \ \sum_{j=1}^{n} w_j = 1 \tag{11}$$

Step 4: Construct the weighted matrix $Z$ for the NQPF development level index system.

$$Z = \left(x'_{ij} \times W_j\right)_{m \times n} \tag{12}$$

Step 5: Identify the optimal plan $Z_j^+$ and the worst plan $Z_j^-$. In this step, the optimal plan $Z_j^+$ and the worst plan $Z_j^-$ are determined. These plans represent the best and worst possible values for each indicator, respectively.

$$D_j^+ = \sqrt{\sum_{j=1}^{m}\left(Z_{ij} - Z_j^+\right)^2} \tag{13}$$

$$D_j^- = \sqrt{\sum_{j=1}^{m}\left(Z_{ij} - Z_j^-\right)^2} \tag{14}$$

Step 6: Calculate the relative closeness $C_i$ of each measurement plan to the ideal plan.

$$C_i = \frac{D_i^-}{D_i^+ - D_i^-} \tag{15}$$

The larger the $C_i$ value, the higher the level of NQPF development in city $i$; conversely, the smaller the $C_i$ value, the lower the level of NQPF development in city $i$.

## Supporting information

**S1 Data. Original data.**

## Author contributions

**Conceptualization:** Kun Cheng, Jianhui Yin, Feiyan Wang, Min Wang.

**Data curation:** Jianhui Yin, Feiyan Wang, Min Wang.

**Formal analysis:** Kun Cheng, Min Wang.

**Funding acquisition:** Kun Cheng, Feiyan Wang.

**Investigation:** Kun Cheng, Jianhui Yin, Feiyan Wang, Min Wang.

**Methodology:** Jianhui Yin, Feiyan Wang, Min Wang.

**Project administration:** Kun Cheng.

**Resources:** Kun Cheng, Jianhui Yin, Feiyan Wang, Min Wang.

**Software:** Kun Cheng, Jianhui Yin, Feiyan Wang, Min Wang.

**Supervision:** Kun Cheng.

**Validation:** Feiyan Wang.

**Visualization:** Jianhui Yin, Min Wang.

**Writing – original draft:** Kun Cheng, Jianhui Yin, Feiyan Wang, Min Wang.

**Writing – review & editing:** Kun Cheng, Jianhui Yin, Feiyan Wang, Min Wang.

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
