## [Decision Letter · Decision Letter 0]

29 Nov 2024

PONE-D-24-43987The Impact Pathway of New Quality Productive Forces on Regional Green Technology Innovation: A Spatial Mediation Effect Based on Intellectual Property ProtectionPLOS ONE

Dear Dr. Yin,

Thank you for submitting your manuscript to PLOS ONE. After careful consideration, we feel that it has merit but does not fully meet PLOS ONE’s publication criteria as it currently stands. Therefore, we invite you to submit a revised version of the manuscript that addresses the points raised during the review process.

We look forward to receiving your revised manuscript.

Kind regards,

Pedro Ribeiro Mucharreira, Ph.D.

Academic Editor

PLOS ONE

**Journal Requirements:**

This work was supported in part by the Major Research Projects of Humanities and Social Sciences in Universities in Anhui Province of China under Grant 2024AH040285.

3. In the online submission form, you indicated that Data will be made available on request.

4. We note that Figures 3 and 4 in your submission contain map images which may be copyrighted. All PLOS content is published under the Creative Commons Attribution License (CC BY 4.0), which means that the manuscript, images, and Supporting Information files will be freely available online, and any third party is permitted to access, download, copy, distribute, and use these materials in any way, even commercially, with proper attribution. For these reasons, we cannot publish previously copyrighted maps or satellite images created using proprietary data, such as Google software (Google Maps, Street View, and Earth). For more information, see our copyright guidelines: http://journals.plos.org/plosone/s/licenses-and-copyright.

We require you to either present written permission from the copyright holder to publish these figures specifically under the CC BY 4.0 license, or remove the figures from your submission:

a. You may seek permission from the original copyright holder of Figures 3 and 4 to publish the content specifically under the CC BY 4.0 license.  

5. Please remove your figures from within your manuscript file, leaving only the individual TIFF/EPS image files, uploaded separately. These will be automatically included in the reviewers’ PDF.

Reviewers' comments:

Reviewer's Responses to Questions

**Comments to the Author**

1. Is the manuscript technically sound, and do the data support the conclusions?

Reviewer #1: Yes

Reviewer #2: Yes

2. Has the statistical analysis been performed appropriately and rigorously? 

Reviewer #1: Yes

Reviewer #2: No

3. Have the authors made all data underlying the findings in their manuscript fully available?

Reviewer #1: Yes

Reviewer #2: Yes

4. Is the manuscript presented in an intelligible fashion and written in standard English?

Reviewer #1: Yes

Reviewer #2: No

5. Review Comments to the Author

**Reviewer #1: ** The title reflects the purpose of the research. The summary contains the purpose of the research, the data and methodology used and the main results. The introduction should be improved by giving an overview of the indicators' dynamics from 2011 to 2022. It is important to note below the tables what the values in brackets or *** represent for general public. The conclusions present the practical implications of the results of the empirical analysis.

**Reviewer #2: ** I am honored to review your manuscript, but the following modifications are required to meet the publishing standards.

Firstly, both the explanatory variable and the dependent variable should undergo Moran's index testing.

Secondly, please keep all regression coefficients to only 3 decimal places.

Thirdly, in the third column of Table 5, the regression of spatial effects is not significant, and nQPF does not have spatial spillover effects on green technology innovation. Therefore, even if the quadratic term of nQPF is significant, it cannot prove its inverted "u" characteristic.

Fourthly, the literature review and hypothesis proposal sections need to be revised to better introduce the mediating variable IPP.

Fifth, the variables in the article need to be explained more carefully, including definitions, sources, and reasons for selecting control variables.

6. PLOS authors have the option to publish the peer review history of their article (what does this mean? ). If published, this will include your full peer review and any attached files.

**Do you want your identity to be public for this peer review?** For information about this choice, including consent withdrawal, please see our Privacy Policy .

Reviewer #1: No

Reviewer #2: No

---

## [Author Response · Author response to Decision Letter 1]

7 Jan 2025

We are grateful for the opportunity to submit a revised draft of the manuscript " The Impact Pathway of New Quality Productive Forces on Regional Green Technology Innovation: A Spatial Mediation Effect Based on Intellectual Property Protection" for publication in the PLOS ONE. We appreciate the time and effort that editors and the reviewers dedicated to providing feedback on our manuscript and are grateful for the insightful comments and valuable improvements to our paper. We have incorporated all of the suggestions made by the reviewers. Please see below, for a point-by-point response to the re- viewer’ comments and concerns. All section numbers refer to the revised manuscript file with tracked changes.

---

## [Editor Report · Decision Letter 1]

9 Feb 2025

The Impact Pathway of New Quality Productive Forces on Regional Green Technology Innovation: A Spatial Mediation Effect Based on Intellectual Property Protection

PONE-D-24-43987R1

Dear Dr.Jianhui Yin,

We’re pleased to inform you that your manuscript has been judged scientifically suitable for publication and will be formally accepted for publication once it meets all outstanding technical requirements.

Kind regards,

Pedro Ribeiro Mucharreira, Ph.D.

Academic Editor

PLOS ONE
---

## [Editor Report · Acceptance letter]

PONE-D-24-43987R1

PLOS ONE

Dear Dr. Yin,

I'm pleased to inform you that your manuscript has been deemed suitable for publication in PLOS ONE. Congratulations! Your manuscript is now being handed over to our production team.

Kind regards,

on behalf of

Dr. Pedro Ribeiro Mucharreira

Academic Editor

PLOS ONE